# Role of the Cholinergic Anti-Inflammatory Reflex in Central Nervous System Diseases

**DOI:** 10.3390/ijms222413427

**Published:** 2021-12-14

**Authors:** Ivan Emmanuel Ramos-Martínez, María Carmen Rodríguez, Marco Cerbón, Juan Carlos Ramos-Martínez, Edgar Gustavo Ramos-Martínez

**Affiliations:** 1Glycobiology, Cell Growth and Tissue Repair Research Unit (Gly-CRRET), Université Paris Est Créteil (UPEC), 94010 Créteil, France; iramos.martinez88@gmail.com; 2Centro de Investigación Sobre Enfermedades Infecciosas, Instituto Nacional de Salud Pública, SSA, Morelos 62100, Mexico; mrodri@insp.mx; 3Unidad de Investigación en Reproducción Humana, Instituto Nacional de Perinatología-Facultad de Química, Universidad Nacional Autónoma de México, Ciudad de México 04510, Mexico; 4Cardiology Department, Hospital General Regional Lic. Ignacio Garcia Tellez IMSS, Yucatán 97150, Mexico; DR.JUANCARLOSRAMOS@hotmail.com; 5Escuela de Ciencias, Universidad Autónoma Benito Juárez de Oaxaca, Oaxaca 68120, Mexico; 6Instituto de Cómputo Aplicado en Ciencias, Oaxaca 68044, Mexico

**Keywords:** neuroimmunomodulation, vagus nerve stimulation, nicotine receptor, Alzheimer’s disease, Parkinson’s disease, experimental autoimmune encephalomyelitis

## Abstract

In several central nervous system diseases, it has been reported that inflammation may be related to the etiologic process, therefore, therapeutic strategies are being implemented to control inflammation. As the nervous system and the immune system maintain close bidirectional communication in physiological and pathological conditions, the modulation of inflammation through the cholinergic anti-inflammatory reflex has been proposed. In this review, we summarized the evidence supporting chemical stimulation with cholinergic agonists and vagus nerve stimulation as therapeutic strategies in the treatment of various central nervous system pathologies, and their effect on inflammation.

## 1. Introduction

Different mechanisms indicating bidirectional communication between the nervous system and the immune system have been reported. This communication is possible because common receptors and ligands are expressed in cells of both systems. Through these receptors, the nervous system has a regulatory role on the immune system, defined as neuroimmunomodulation [1]. Neuroimmunomodulation can occur by direct or indirect mechanisms. The first mechanisms include neurotransmitters (e.g., norepinephrine and acetylcholine), which act on cells from the immune system through their receptors (adrenergic receptors and nicotinic acetylcholine receptors) [2,3,4]. In indirect mechanisms, specific reflex stimuli act on blood vessel cells to express chemokines and facilitate the entry of immune cells into specific tissues [2].

Several nervous reflexes have been shown to induce immune system responses [2], being the cholinergic anti-inflammatory reflex, mediated by the vagus nerve the most studied. Stimulation of the vagus nerve controls inflammation at the peripheral level, but also in the central nervous system through α7 nicotinic acetylcholine receptors (α7nAChR). This evidences prompted to employ of the vagus nerve electrical stimulation and modulation with α7nAChR agonists, as alternative therapies for neurological and neuropsychiatric disorders [5]. Therefore, in this review, we focus on the modulation of the anti-inflammatory reflex through the vagus nerve stimulation and pharmacological approach, as an alternative therapy for central nervous system diseases.

## 2. Inflammatory Reflex: Cholinergic Anti-Inflammatory Pathway

A nervous reflex is an involuntary and fixed response that originates from an afferent signal; it is said to be fixed because it cannot change or adapt to circumstances. Many nervous simple and complex reflex functions have been described, e.g., simple as the axon reflex, or complex, such as somatic and visceral reflexes [6]. Interestingly, the reflex functions play a key role in the regulation of the immune system [6].

Nerves innervate the skin and visceral organs, so it is not surprising that they have a major function in the detection of infections and injuries [6]. Neurons can sense pathogens or harmful molecules by expressing pattern recognition receptors (PPRs), such as toll-like receptors (TLRs) 1–5, 7, and 9 [7,8], NOD-like receptors (NLRP 1, 3, and 4), and inflammasome AIM2 [9,10,11]. In addition, they express chemokine receptors [12,13] and cytokines [14]. This allows neurons to sense tissues for inflammatory signals, and to participate in the regulation of the immune system through reflexes. Several reflexes acting on the immune system have been described and have been reviewed by Kamimura et al. (2020) [2]. In the next sections, we describe the molecular and cellular mechanism of the cholinergic anti-inflammatory reflex, which is the most widely studied.

Pioneering work by Bernik et al. (2002), showed that signals mediated by the vagus nerve exert a crucial role in the regulation of inflammation. In this study it was shown that in rats subjected to endotoxemia, hypotension was reduced and the production of Tumor Necrosis Factor α (TNFα) in the heart, liver, spleen, and other organs, was blocked by intracerebroventricular injections of CNI-1493 (a p38 MAPK (mitogen-activated protein kinase) inhibitor, derived from guanylhydrazone, which blocks cytokine secretion). After surgical vagotomy, treatment with CNI-1493 did not reduce hypertension or TNFα production [15]. Furthermore, when electrical stimulation of the vagus nerve was performed, hypotension and TNFα production were also reduced [15,16]. Therefore, it was concluded that the vagus nerve could regulate cytokine secretion in the spleen and other organs, a phenomenon known as the anti-inflammatory reflex or cholinergic anti-inflammatory pathway [16].

In the anti-inflammatory reflex, the afferent vagus nerve responds to inflammatory mediators sending signals to the tractus solitarius nucleus of the brainstem. Once the signal has reached the brainstem, a response signal originates in the dorsal motor nucleus of the vagus that travels through the efferent vagus nerve and generates the anti-inflammatory response [6,17].

The efferent vagus nerve transmits a signal to the splenic nerve with subsequent release of noradrenaline (NA) [18]. A subset of memory CD4+ T lymphocytes that express the beta-2 adrenaline receptor (β2AR), capture noradrenaline, and release acetylcholine [19]. The CD4+ T lymphocytes express the enzyme choline acetyltransferase, responsible for acetylcholine synthesis, and are found in the spleen, lymph nodes, and Peyer’s plaques [19]. Acetylcholine, in turn, acts on macrophages expressing the α7nAChR receptor thereby inhibiting lipopolysaccharide (LPS)-induced production of proinflammatory cytokines [20]. Acetylcholine can inhibit the production of TNFα, IL (interleukin)-1β, IL-6, and IL-18, but not IL-10, during lethal endotoxemia in rats [16]. In LPS-injected rats, muscarine administration centrally affects TNF levels, however, when administered peripherally it lacks this effect, suggesting that signaling through muscarinic receptors in macrophages and other peripheral cells do not affect TNF levels [21].

Suppression of inflammatory cytokine secretion in macrophages by acetylcholine is due to the blockage of nuclear factor-kB (NFκB) translocation [22] and activation of the Janus kinase 2/signal transducer and activator of transcription 3 (JAK2/STAT3) pathway [23]. Another mechanism that prevents the production of proinflammatory cytokines in macrophages is related to the inactivation of the NACHT, LRR, and PYD domains-containing protein 3 (NLRP3) inflammasome [6]. Acetylcholine can interact with the α7nAChR in the mitochondria of macrophages, which reduces mitochondrial damage induced by hydrogen oxide and calcium, and blocks the release of mitochondrial DNA. This prevents the activation of NLRP3 inflammasome [24].

The cholinergic anti-inflammatory reflex also has other effects on the immune system. For example, electrical stimulation of the vagus nerve and nicotine administration stopped B cell migration and decreases antibody secretion in BALB/c mice injected intravenously with a suspension of Streptococcus pneumoniae strains [25]. In a model of hypertension, the induced signals travel through the vagus and splenic nerves to stimulate the activation and migration of T cells, which migrate to target organs and participate in blood pressure regulation [26]. Furthermore, the anti-inflammatory reflex reduces the expression of adhesion molecules on endothelial cells by inhibiting NF-kB entry into the nucleus [27]. Additionally, it decreases leukocyte migration by suppressing F-actin polymerization and reducing membrane expression of CD11b on neutrophils [28] (Figure 1).

The anti-inflammatory reflex also participates in the resolution of inflammation by stimulating the release of lipid mediators, known as pro-resolving mediators (SPMs), from cells of the immune system [29,30]. SPMs include resolvins, protectins, lipoxins, and maresins. These molecules trigger potent anti-inflammatory and pro-resolving mechanisms and enhance microbial clearance. Some of the functions they perform include limiting polymorphonuclear leukocyte infiltration and tissue damage and enhancing macrophage phagocytosis. For a more extensive review of these molecules, the work of Serhan et al. (2015) can be consulted [31].

In an Escherichia coli infection model, dissection of the right vagus nerve delayed the resolution of infection, which was related to a peritoneal decrease in the concentration of protective immunoresolvent resolvin 1 (PCTR1). The vagus nerve promotes PCTR1 production in peritoneal group 3 innate lymphoid cells (ILC3s), through the release of acetylcholine. Moreover, PCTR1 produced by ILC3s autoregulates its production in peritoneal macrophages and favors the resolution of infection [29]. In another study, it was reported that the vagus nerve favors the local expression of netrin-1, a molecule involved in the resolution of inflammation by decreasing the recruitment of nuclear polymorphs, reducing inflammatory mediators, and stimulating the production of resolvins, protectins, and lipoxins [30]. These studies demonstrate the role the anti-inflammatory reflex has in the resolution of infection.

Although the anti-inflammatory reflex was initially related to acetylcholine signaling through nicotinic acetylcholine receptors, signaling through muscarinic receptors also has effects on the immune system. For example, in antigen-presenting cells (APCs), signaling through muscarinic receptors contributes to the maintenance of regulatory T cells in the intestine (Figure 1). Vagotomy of the hepatic afferent branch of the vagus nerve reduces the number of regulatory T cells by decreasing aldehyde dehydrogenase expression and retinoic acid synthesis in intestinal antigen-presenting cells [32]. Furthermore, in the prosencephalon, acetylcholine signaling on mAChR M1 receptors (M1 muscarinic acetylcholine receptor) decreases serum TNFα levels in mice with endotoxemia [33]. Next, we review the clinical applications of anti-inflammatory reflex modulation.

## 3. Clinical Applications of Inflammatory Reflex Modulation

Pharmacological and electrical stimulation of the vagus nerve has been investigated in a wide range of diseases such as autoimmune, cardiovascular system, respiratory system, and infectious diseases. Vagus nerve electrical stimulation (VNS) was approved by the US Food & Drug Administration (FDA) for the treatment of drug-resistant epilepsy, depression, and migraine [34]. In addition, there are clinical trials for the treatment of Crohn’s disease [35], rheumatoid arthritis [36], SARS-Cov2 infection [37], and systemic lupus erythematosus with VNS [38]. Recently, an experimental study also showed that the inflammatory response depends on VNS parameters. When healthy mice are subjected to different parameters of pulse width, pulse amplitude, and frequency during four minutes of VNS, the inflammatory response can vary, from an inflammatory profile characterized by an increase in TNF-α to an anti-inflammatory response with an elevation of IL-10. The authors highlight the importance of standardization and optimization of parameters during VNS treatment [39].

In the case of cardiovascular diseases, increased vagal activity has been observed to reduce vascular inflammation and cardiovascular risk in patients and animal models [40,41]. Thus, noninvasive electrical stimulation for the treatment of cardiovascular disorders offers a very active new field of research [42]. Moreover, in patients with rheumatoid arthritis treated with a device that electronically stimulates the left cervical vagus nerve, TNFα production and disease severity were reduced [36]. Electrical stimulation of the vagus nerve has been proposed as replacement therapy for pharmacological therapies in rheumatoid arthritis [43].

Chemical modulation of the inflammatory reflex is an attractive option that has also been explored. Treatment with pyridostigmine, an acetylcholinesterase inhibitor, has been approved for clinical use in cases of myasthenia gravis [44]. Furthermore, in a model of LPS-induced Acute respiratory distress syndrome (ARDS), treatment with pyridostigmine reduced the number of macrophages and lymphocytes in bronchoalveolar lavage fluid and suppressed levels of TNFα, IL-1β, IL-6, and IFN-γ [45]. A phase 2/3 clinical trial is currently underway to test the effects of anti-inflammatory reflex modulation in Severe acute respiratory syndrome coronavirus 2 (SARS-CoV-2) infection, with the use of pyridostigmine as adjuvant therapy, along with baseline therapy, reducing the use of invasive mechanical ventilation and death [46].

Next, we review the involvement of the anti-inflammatory reflex in central nervous system diseases and the main advances in chemical modulation with α7 nAChR agonists and VNS.

## 4. Role of the Anti-Inflammatory Reflex and Main Advances in Chemical Modulation with α7 nAChR Agonists and VNS in Central Nervous System Diseases

### 4.1. Experimental Autoimmune Encephalomyelitis and Multiple Sclerosis

Experimental autoimmune encephalomyelitis (EAE), the experimental model of multiple sclerosis, is characterized by infiltration of cells of the innate and adaptive immune system into the CNS due to the loss of blood-brain barrier integrity [47]. Cells such as monocytes, macrophages, and endothelial cells produce IL-1β, granulocyte-monocyte colony-stimulating factor (GM-CSF), and granulocyte colony-stimulating factor (G-CSF) [48], which promote the differentiation of infiltrating monocytes into antigen-presenting cells. In addition, activated microglia cells also participate as antigen-presenting cells [49]. Antigen-presenting cells secrete IL-12p70, IL-6, and transforming growth factor-beta (TGF-β), which favor the differentiation of CD4+ T lymphocytes to Th1 and Th17 subpopulations [50]. Once differentiated, Th1 and Th17 lymphocytes enter the CNS where they induce the adaptive immune response characteristic of EAE [51,52].

The cholinergic anti-inflammatory pathway has received attention as a possible pathway for neuro-immunomodulation of inflammation in EAE and multiple sclerosis [53,54,55]. Among the data supporting a relationship between defects in the cholinergic pathway and inflammation, are low serum acetylcholine levels in relapsing multiple sclerosis patients, probably due to an increase in the activity of hydrolyzing enzymes such as, acetylcholinesterase (AChE) and butyrylcholinesterase (BChE) enzymes, that degrade acetylcholine. In addition, the decrease in acetylcholine levels is related to high levels of expression of inflammatory cytokines such as TNFα, IL-12/IL-23p40, and IL-18 [56]. The relationship between alterations in the cholinergic system and inflammation in multiple sclerosis has recently been reviewed by Gatta (2020) [57].

Modulation of the cholinergic anti-inflammatory pathway, using α7 nAChR agonists, produces different effects on the immune response and activity in the EAE model [58,59]. For example, treatment with the synthetic agonist GAT107 (the (+)-enantiomer of racemic 4-(4-bromophenyl)-3a,4,5,9b-tetrahydro-3H-cyclopenta[c] quinoline-8-sulfonamide, is a strong positive allosteric modulator of α7 nAChR) reduces disease severity, T-cell proliferation, inflammatory cytokine synthesis, and anti-MOG35-55 antibody production, in addition to increased IL-10 synthesis [59]. On the other hand, the transfer of acetylcholine-producing NK cells to the brain ventricles can reduce brain damage, monocyte infiltration, and immune responses in the CNS of EAE mice [60].

Agonists of the α7 nAChR such as nicotine and small heat shock proteins, reduce the secretion of proinflammatory cytokines such as TNFα, IL-1β, IL-6, and IL-18 and ameliorate clinical manifestations in EAE. Here it is interesting to note that the therapeutic activity of the small heat shock proteins depends on the property of forming amyloid fibrils, this was demonstrated by making point mutations that affect fibril formation and the therapeutic effect disappears [61]. These agonists reduce clinical paralysis and inflammation by activating the STAT3 pathway and inhibiting the mTorC1 pathway in regulatory macrophages that limit T lymphocyte proliferation [62]. The presence of B lymphocytes was required for amyloid fibrils and small heat shock proteins to exert their protective effect, possibly because a population of B1-a lymphocytes releases IL-10 upon stimulation with amyloid fibrils. Furthermore, B lymphocytes had been previously shown to be essential for ameliorating inflammatory brain disease [63].

The modulatory effect of α7 nAChR agonists has also been described in cells of the innate immune system. For example, nicotine inhibits monocyte and neutrophil infiltration in the CNS of mice developing EAE, by reducing the expression of CCL2 and CXCL2 chemokines. This effect was dependent on signaling through α7 nAChR and α9 nAChR receptors [64]. Furthermore, stimulation through the α7 nAChR inhibits NLRP 3 inflammasome activation in monocytes and microglia from EAE mice, which reduced LPS-stimulated IL-1β and IL-18 release [65]. In peripheral blood mononuclear cell cultures from multiple sclerosis patients, nicotine stimulated the expression of α7 nAChR and decreased the expression of IL-1β and IL-17 [66]. However, it has been suggested that the effects observed upon stimulation of α7 nAChR receptors occur by inducing the desensitized state of the receptor and not by opening ion channels. This is because treatment with silent α7 nAChR agonists (molecules that selectively induce the desensitized state of nicotinic acetylcholine receptors and do not induce the opening of ion channels) reduces the clinical manifestations of EAE and the number of monocytes infiltrating the CNS. In addition, they reduce the number of monocytes and macrophages derived from mouse bone marrow from EAE C57BL/6J mice after treatment with an α7 nAChR-selective silent agonist, 1-ethyl-4-(3-(bromo)phenyl)piperazine (m-bromo PEP) besides a decrease the production of inflammatory cytokines, in cultures [67].

Previous results support an anti-inflammatory effect for α7 nAChR agonists and a possible beneficial effect of anti-inflammatory reflex stimulation in EAE. However, in mice lacking the α7 nAChR receptor and EAE induced, no differences were found in disease progression, clinical score, and lesion composition (number of CD3+ T lymphocytes, macrophages, and activated microglia) compared with mice that expressed the receptor. In addition, the expression of the α7 nAChR receptor was not associated with a decrease in TNFα, IL-1β, and IL-18 [68]. Another study showed that acetylcholine suppression, by left cervical vagotomy, improved clinical symptoms and decreased IL-6, IL-4, IL-17, and IFNγ in EAE. A possible mechanism for this effect is the regulatory role of acetylcholine on T lymphocyte proliferation and differentiation, as it has been shown in vasectomized mice, which presented a decrease in the percentage of Th1 lymphocytes [69].

In conclusion, the use of α7 nAChR agonists in EAE has shown interesting results in preclinical studies; however, studies are needed to clarify the advantages of an inhibitory or stimulatory therapy of the cholinergic anti-inflammatory pathway. Regarding VNS, controlled trials are required to evaluate the improvement in clinical manifestations and their relationship with inflammatory parameters.

### 4.2. Alzheimer’s Disease

Alzheimer’s disease (AD) is the most common neurodegenerative disorder in older adults. There is no effective treatment and is characterized by memory impairment. The histopathological lesions that identify AD are amyloid plaques, neurofibrillary tangles, and neuroinflammation [70]. Neuroinflammation is manifested by activated astrocytes and microglia surrounding amyloid plaques, in addition to elevated levels of inflammatory cytokines such as IL-1β, IL-6, and TNF-α in the brain [71,72].

Some clinical trials evaluating the effectiveness of nonsteroidal anti-inflammatory drugs (naproxen and celecoxib) and a specific TNF-α inhibitor (Etanercept) did not show improvements in cognitive symptoms or disease progression [73,74,75]. However, therapeutic approaches based on the modulation of inflammation are relevant and recently were reviewed by Sanchez-Sarasua et al. (2020). In this review, the authors addressed the translational evidence of the main anti-inflammatory strategies in AD and propose that treatment of neuroinflammation should contemplate multiple approaches [76]. As neuroinflammation and cholinergic deficit are processes involved in AD [77], we review the main findings of modulation of inflammation through the cholinergic anti-inflammatory pathway in this disease.

In AD, the interest of the cholinergic pathway is not only related to the regulation of inflammation, but also its effect on cognition and neuroprotection. Stimulus through the α7 nAChR regulates functions related to cognition, neurotransmitter release, pain, synapse plasticity, and neuronal protection [78]. In AD, new therapeutic targets have been explored that focus on the modulation of the cholinergic pathway by administration of acetylcholinesterases inhibitors, α7 nAChR agonists, and vagus nerve stimulation. Acetylcholinesterase inhibitors are among the drugs of the first choice for the treatment of AD [79]. Treatment of AD patients with acetylcholinesterase inhibitors (donepezil) has reduced the secretion of IFN-γ, TNF-α, IL-1β, and IL-6 by peripheral blood mononuclear cells [71] and favored the secretion of monocyte chemotactic protein-1 (MCP-1) and IL-4 [80]. However, another study reported no significant effects of acetylcholinesterase inhibitors on plasma cytokines in patients with AD [81].

Nicotine administration as an α7 nAChR agonist, ameliorates cognitive impairment in AD [82,83], although this has been mainly related to its effect on learning and memory processes and not to its effect on inflammation [82,84]. Recently, Alhowail (2021) has reviewed the benefits of nicotine on memory and cognition and postulates that nicotine regulates these processes through the stimulation of the phosphoinositide 3-kinase/AKT pathway [82]. Another α7 nAChR agonist, EVP-6124, showed functional and cognitive improvements in patients with mild to moderate AD in phase 2 clinical trials [85]. However, the drug was discontinued due to its gastrointestinal side effects [79].

Regarding the stimulation vagus nerve stimulation (VNS), various studies show that VNS can modulate cognitive functions in animal models and aim to use VNS as a therapy to improve memory and learning in patients with AD. Among the first studies in laboratory animals, Clark et al. (1998), suggest that VNS (0.5-ms biphasic pulses; 20.0 Hz; 30 s; 0.2, 0.4, or 0.8 mA) can increase memory in rats trained on an inhibitory-avoidance task [86]. The mechanism by which VNS can stimulate memory is not clear, however, a study in anesthetized rats showed that VNS has a direct effect on the hippocampus inducing an increase in synaptic transmission in perforant path projection to CA3 (PP-CA3), involved in the cellular mechanism of learning and memory. Additionally, they showed that lesion in locus coeruleus (LC) and the administration of timolol prevented this effect, suggesting that VNS can stimulate PP-CA3 of the hippocampus via LC and β-adrenergic receptors [87]. Another proposed mechanism is that VNS can induce hippocampal formation theta rhythm, involved in memory encoding and spatial memory. Additionally, they showed that medial septal injection of procaine abolished VNS-induced HPC theta rhythm, suggesting that medial septal is the best way to bring the vagal stimulus to the formation of the hippocampus [88].

In AD animal models, the effect of VNS has been evaluated, obtaining interesting results. For example, amyloid precursor protein/presenilin-1 double-transgenic mouse (APP/PS1) revealed that Non-invasive vagus nerve stimulation (nVNS) induces a change in microglial morphology to a neuroprotective phenotype on APP/PS1 mice. Untreated APP/PS1 mice showed a neurodestructive effect in microglial morphology, and wild-type mice did not show a negative effect on microglial morphology or obvious side effects after nVNS treatment [89]. In another study, aged rats with postoperative cognitive dysfunction (POCD) were treated with auricular vagus nerve stimulation (aVNS). The results showed a decrease in the levels of IL-1β and TNF-α, and the expression of tau phosphorylation and amyloid β42 in the hippocampus. Additionally, Morris’s water tests showed less spatial memory impairment. The authors conclude that aVNS exerted a neuroprotective effect and attenuated neuroinflammation [90].

Several trials suggested that VNS may be beneficial for various cognitive functions in humans. In a randomized double-blind clinical trial using VNS as a treatment for epilepsy, Clark et al. (1999) evaluated verbal learning in humans. The researchers noted that VNS enhanced word recognition performance compared to sham [91]. Moreover, a trial that recruited 17 patients, with probable AD, were treated with VNS. After one year of treatment, patients showed improvement or no deterioration in baseline test scores; ADAS-cog (41.2%) and MMSE (70.6%). The authors concluded that VNS can be used as long-term therapy and only transient and mild adverse effects were found [92,93]. In a more recent study, the effect of transcutaneous vagus nerve stimulation (tVNS) on associative memory from 30 healthy older individuals was evaluated. Participants performed a face-name association memory task where the researchers observed an enhanced number of hits in the memory task and few side effects after tVNS treatment [94].

Therefore, we suggest that cholinergic signaling in AD is involved both in the regulation of memory and cognition, and may also have important effects on inflammation as shown by studies in animal models and patients [33]. So far there are interesting studies in animal models that suggest the efficacy of VNS in memory improvement. However, in the case of clinical trials in humans, studies with larger samples and a more rigorous experimental design are necessary. Table 1 shows the VNS parameters tested in clinical trials in some diseases of the central nervous system. The use of non-invasive VNS is promising (tVNS and aVNS) as it will allow greater acceptance in future studies.

### 4.3. Parkinson’s Disease

Parkinson’s disease (PD) is a neurodegenerative disorder from dopaminergic neurons in the substantia nigra pars compacta [100]. In PD, the loss of dopaminergic neurons has been linked to neuroinflammation and activation of microglia, lymphocyte infiltration, and elevated levels of proinflammatory cytokines [101]. Neuroinflammation has been observed in many PD models, such as those administered with the neurotoxin 6-hydroxydopamine, 1-methyl-4-phenyl-1,2,3,6-tetrahydropyridine (MPTP), or misfolded α-synuclein [102]. Epidemiological studies and in vivo models have suggested that anti-inflammatory therapies can modify PD progression [102,103].

The cholinergic system has received attention as a therapeutic target in PD, not only because of its effect on neuroinflammation, but also because it modulates dopamine release, protects against degeneration of dopaminergic neurons, and rebalances direct and indirect signaling pathways in the striatum [100]. Liu (2020) reviewed the cholinergic pathway as a therapeutic target in PD emphasizing its effect on striatum signaling [100]. Next, we review the effect of cholinergic pathway modulation on PD-associated neuroinflammation.

In a cellular model of PD, it was reported that agonists of the α7 nAChRs inhibit cell death. In this study neuroblastoma cells (SH-SY5Y line) were treated with 1-methyl-4-phenylpyridinium (MPP), a metabolite of MPTP, which induces cell death and secretion of pro-inflammatory cytokines such as TNF-α, IL-1β, and IL-6. When SH-SY5Y cells were coadministered with MPP plus choline or nicotine cell death was inhibited, this protective effect involved activation of the extracellular signal-regulated kinase/p53 (ERK/p53) pathway, and the decreased cleavage of Poly (ADP-ribose) polymerase-1 (PARP-1) and caspase-3; two proteins related to PD and cell apoptosis [104,105]. However, in this study, it was not determined whether nicotine or choline treatment reduced inflammatory cytokine levels which could be another mechanism that reduces cell death [106], as previous studies have shown that agonists of the α7 nAChRs reduce oxidative stress and neuroinflammation [107].

In an animal model with mice deficient in α7 nAChR and treated with MPTP, there was an increased deposition of α-synuclein; a protein whose misfolding and aggregation increases the risk of developing PD [108] and leads to loss of dopaminergic neurons [109]. Nicotine and PNU-282987, a selective alpha7 nicotinic receptor agonist, promoted the elimination of α-synuclein aggregates and inhibited neuronal apoptosis in vitro [109].

Concerning inflammation, PNU-282987 reduced the concentration of IL-1β and TNF-α. Also, it reduced the number of activated microglia and prevented the loss of dopaminergic neurons in MPTP-treated mice [110]. Furthermore, the combined therapy of α7 nAChR and sigma-1 receptor (s1-R) agonists induced partial protection of dopaminergic neurons. The authors propose that this effect is due to the reduction of microglia activation [111].

Regarding the stimulation of the cholinergic pathway by VNS, this procedure has not been approved as a treatment for PD. However, some studies in animal models have shown its potential usefulness. For example, the therapeutic potential of VNS in PD was explored in a model with intrastriatal DSP-4/6-OHDA (N-(2-chloroethyl)-N-ethyl-2-bromobenzylamine / 6-hydroxydopamine) in rats, where one group was treated with VNS for ten days and the other with sham treatment. The VNS group showed improvements in locomotion and increased tyrosine hydroxylase (TH) expression. In addition, decreased expression of GFAP and Iba-1 in glia cells was observed, which are overexpressed in neuroinflammatory processes [112].

Similar results were observed in another study where 6-hydroxydopamine was administered in the medial forebrain bundle of Wistar rats, followed by VNS treatment. Treatment significantly improved motor deficits and reduced levels of the inflammatory cytokines; tumor necrosis factor-α (TNF-α) and IL-1β. It also increased the number of regulatory T cells while decreasing Th-17 cells [113]. In the third study with 6-hydroxydopamine in rat striatum, different intensities of VNS stimulation (0.1, 0.25, 0.5, and 1 mA) were evaluated. Intensities of 0.25–0.5 mA improved movement impairment and reduced astrocyte and microglia activation. Intensities of 0.1 and 1 mA showed no improvement [114]. In both cases, the authors attribute an anti-inflammatory and neuroprotective effect to VNS.

In 2019, Mondal et al. conducted an observational, open-label, pilot study where they explored the effect of nVNS (non-invasive VNS) in patients with PD-related gait disorder and found improvements in gait and reduced gait freezing [115]. Mondal et al. extended the study in a randomized, double-blind, sham-controlled crossover trial with 33 patients with PD-related gait disorder, who were divided into two groups; one treated with nVNS and the other with sham stimulation. The researchers observed improvement in gait parameters such as gait speed, support time, and step length, compared to sham. This improvement is accompanied by decreased levels of TNF-α and glutathione in serum, however, in IL-6 levels there was no significant change. The authors propose that the beneficial effect of nVNS may be related to the anti-inflammatory effect [99].

In conclusion, epidemiological and model studies indicate a potential use of chemical agonists and VNS in the treatment of PD. However, whether stimulation or inhibition of nAChRs will be beneficial for patients, needs to be clarified, as inhibition of nAChRs reduces motor deficits [116]. Clinical studies evaluating the effect of α7 nAChR agonists and VNS on inflammation in PD patients are also required. Currently, there is only one ongoing clinical trial evaluating the role of VNS on motor and cognitive function in PD patients (NCT04157621) but does not evaluate the effect on inflammation.

### 4.4. Epilepsy

Neuroinflammation has a key role in the development of epilepsy [117,118]. This is supported by the fact that chronic inflammation in autoimmune diseases or recurrent infections may be a cause for the development of epilepsy [119], as patients with autoimmune diseases are more than four times more likely to develop this disorder [117]. Furthermore, TNF-α and IL-1β are found to be increased in the kainic acid-induced epilepsy model, and seizure susceptibility correlates with microglia development in the hippocampus [120]. High-mobility box 1 group protein and Toll-like receptor 4 (TLR4) antagonists delay seizures in murine models, and TLR4-deficient mice are resistant to kainic acid-induced seizures [121].

A connection between epilepsy, neuroinflammation, and the cholinergic pathway has been suggested [122,123]. This is based on observations where patients with untreatable secondary epilepsy express reduced levels of α7 nAChR in epileptogenic foci tissues [124]. Also, acetylcholine is increased in status epilepticus, in chemically or electrically induced epilepsy models [125]. Furthermore, increased expression of acetylcholinesterases was reported in the hippocampus, microglia, and endothelial cells in mice where status epilepticus was induced with pilocarpine [126]. The increase in acetylcholinesterases was related to an increase in inflammatory cytokines (IL-1β, IL-12, and TNFα), activation of microglia, and development of epilepsy. This suggests a possible regulation of inflammation in epilepsy through the cholinergic pathway.

Many studies have shown an anti-inflammatory effect and decreased seizures in epilepsy models treated with α7 nAChR agonists. For example, treatment with choline chloride decreases seizure severity, depression, and memory deficit in the murine model of epilepsy caused by postpentylenetetrazole administration [127]. Furthermore, JN403 ((S)-(1-aza-bicyclo[2.2.2]oct-3-yl) carbamic acid (S)-1-(2-fluorophenyl) ethyl ester a selective α7 nAChR agonist) decreases epileptic seizures in DBA/2 mice, a model of the audiogenic seizure [128]. Nicotine decreases seizure susceptibility in the Chat-Mecp2-/y mouse, a Rett syndrome model [129], and cytisine reduces seizures, hippocampal damage, and synaptic remodeling in pilocarpine-induced epileptic rats. In addition, cytisine increased the expression levels of ACh and α7 nAChR in the hippocampus [130].

Some studies have reported that α7 nAChR stimulation does not affect inflammation and seizures. For example, treatment with methyllycaconitine citrate, an α7 nAChR antagonist, increased susceptibility to pilocarpine-induced seizures in wild-type mice, but increased α7 nAChR activity with the agonist PNU282987 did not affect seizures [124]. Furthermore, acetylcholine treatment reduced IL-1β and TNF-α levels in cultured brain slices, although this effect was muscarinic, not nicotinic receptor-dependent [126]. Thus, studies comparing the efficacy of different agonists and evaluating possible effects of agonist redundancy are required.

Vagus nerve stimulation (VNS) is a therapeutic option in the treatment of drug-resistant epilepsy, also called refractory epilepsy, and for more than 20 years has demonstrated its efficacy in reducing seizure frequency in pediatric and adult patients. Currently, noninvasive VNS alternatives are being explored and appear promising [131]. However, the mechanisms of action of VNS in epilepsy are unknown. Its effect could be related to the cholinergic anti-inflammatory pathway (CAP) of the inflammatory reflex that can be activated during vagus nerve stimulation.

Majoie et al. explored the effect of VNS treatment on serum levels of the cytokines IL-6, TNF-α, and IL-10 in 11 patients with refractory epilepsy. After 28 weeks of VNS treatment, they found a decrease in IL-6 levels in the positive responders, while levels of IL-10, which has an anti-inflammatory effect, increased. In addition, they observed a decrease in cortisol levels after VNS treatment. The authors conclude that this was a preliminary study and that the number of patients needs to be increased to reach statistical significance in future studies [132]. In a similar study of eight patients with refractory epilepsy and VNS treatment, serum levels of TNF-a, IL-6, and C-reactive protein were measured, and cardiac autonomic function was assessed. However, after three months of VNS treatment, the authors found no significant changes in the measured parameters [95].

In a double-blind randomized design, levels of the cytokines IL-1β, IL-6, and IL-10 were measured in 41 children with refractory epilepsy for 51 weeks and VNS treatment, although no significant changes were found during the study. Interestingly, lower plasma IL-6 levels corresponded with a greater reduction in seizure frequency. The authors attribute the lack of observed changes to the fact that VNS can decrease cytokine levels only at a specific time interval that was not detected by the experimental design used. The analysis of cytokines in CSF was restricted for ethical reasons, so conclusive data could not be obtained in the CNS [96].

Similarly, in another trial, 18F-FDG (2-deoxy-2-[fluorine-18] fluoro-D-glucose) uptake on positron emission tomography (PET/CT) was evaluated to measure arterial wall inflammation in patients with refractory epilepsy and VNS treatment. The authors observed that during VNS treatment 18F-FDG flux was lower compared to when measured one hour later, suggesting that arterial wall inflammation decreases during VNS. The researchers argued that the results were inconclusive because VNS might affect glucose metabolism due to sympathetic activation [133].

Although these studies suggest the importance of the inflammatory reflex in VNS treatment of epilepsy, most had inconclusive results that did not reach statistical significance. More comprehensive experiments in animal models should certainly be considered. There are currently ongoing clinical studies evaluating the effectiveness and safety of VNS in the treatment of epilepsy (NCT03446664, NCT04387435, NCT02076698, NCT04095247), however, these studies only evaluate changes in seizure frequency and severity and changes in heart rate variability. Only one study is evaluating the effect of VNS on cytokine secretion and immune cell activation (NCT03953768). Therefore, future clinical studies should include inflammatory parameters in the results.

### 4.5. Depression

Depression is a psychiatric illness characterized by anhedonia, reduced energy, rumination, impaired cognition, and suicidality. When two or more pharmacological therapies have no beneficial effect on the patient it is known as Treatment-Resistant Depression (TRD) [134]. Depression has an inflammatory component with elevated levels of proinflammatory cytokines, such as tumor necrosis factor-alpha (TNF-α), interleukin (IL)-6, IL-1, and C-reactive protein [135]. In addition, the integrity of the BBB is impaired allowing the entry of surrounding cytokines into the brain and monoaminergic, and glutamatergic neurotransmission is affected [136]. Abnormalities can also be found in the hypothalamic-pituitary-adrenal axis which affects cortisol levels and causes stress [137]. Thus, inflammation may be a potential therapeutic target in the treatment of depression.

The control of inflammation by α7 nAChR has been linked to depression, as knock-out mice for α7 nAChR were observed to develop a depression-like phenotype and exhibit high levels of TNF-α and IL-1β [138]. Furthermore, in an LPS-induced rat model of depression, administration of ketamine and GTS-21([3-(2,4-dimethoxybenzylidene) anabaseine], a selective α7 nAChR agonist) was shown to reduce the levels of IL-1β, IL-6, and TNF-α compared to the group of rats treated with LPS alone. Rats treated with ketamine, or GTS-21, scored better on behavioral tests (sugar water preference rate, numbers of horizontal and vertical movements, and swimming immobility time) than control rats (LPS alone). To prove that the effect was dependent on signaling through the α7 nAChR, rats were treated with ketamine or GTS-21 together with the α7 nAChR antagonist methyllycaconatine (MLA), which inhibited the anti-inflammatory effect [139]. This same effect was observed in PC12 nerve cells where α7 nAChR expression was blocked by RNA interference [139].

Chronic nicotine exposure also has antidepressant effects in murine models, this effect was dependent on α7 nAChR and activation of PI3K/AKT and ERK/CREB pathways. Similar effects were observed for PNU-282987 and GTS-21 [140]. In a murine model of LPS-induced depression, PNU120596 (a positive allosteric modulator of α7 nAChR) administration showed anxiolytic, pro-cognitive, and antidepressant-like effects. The authors related the effect of PNU120596 to a reduction of microglia and astrocyte activation, and reduction of TNF-α and IL-1β concentrations in the hippocampus and prefrontal cortex [141].

On the other hand, it has also been proposed that depression is related to increased activity of the cholinergic system and that stimulation of α7 nAChR, by acetylcholine, mediates depression-like behaviors [142,143,144]. In C57BL/6J mice treated with GTS-21, no antidepressant effect was found, but MLA treatment showed improved scores in the tail suspension and forced swim tests [143]. Furthermore, in male a7 nAChR knockdown mice, a lower depressant effect of treatment with physostigmine, an acetylcholinesterase (AChE) blocker, was observed than in mice that did express the receptor. Finally, in a study of 28 patients with longstanding late-life major depression (LLMD), patients had reduced levels of AChE and BChE activity, suggesting an increase in the cholinergic anti-inflammatory pathway. The decrease in AChE and BChE activity was related to a reduction of glial activation markers in spinal fluid and an increase in plasma IL-6 [145]. These results highlight that the cholinergic system has multiple effects on the mechanisms of depression development and that several mechanisms may be affecting inflammation.

The differences in the reported effects for α7 nAChR may be due to, (1) the model where the effects were evaluated. For example, in models of depression involving LPS with an inflammatory response, α7 nAChR agonists show a favorable response [139,141,146], but in other models signaling through α7 nAChR seems to favor a depressive effect [143,147]. (2) Differences in the behavioral tests performed, as they evaluate different aspects of depression and there is variation in the results between strains and the antidepressant evaluated [148,149,150]. (3) The multiple functions of α7 nAChR, e.g., it is involved in the control of inflammation, neurotransmission, learning, and memory [151,152]. Furthermore, α7 nAChR has been described to form complexes with NMDA glutamate receptors (NMDARs), which modulate mood and depression. Disruption of this complex with a peptide had antidepressant effects [153].

In the case of VNS, the FDA has approved it as an adjuvant treatment in patients with TRD administered in the same period of conventional therapy. In several studies, VNS has shown a positive effect by improving quality of life and decreasing depressive symptoms with few side effects [154,155,156]. However, the parameters and the administration regimen, which requires about six months for optimization, are still under debate [157]. A relationship between the antidepressant effect of VNS and the anti-inflammatory effect has been proposed [158]. However, so far no clinical trial data are determining the effect of VNS on inflammation and cytokine levels.

### 4.6. Migraine

Migraine is a headache disorder that often occurs along with nausea, vomiting, and extreme sensitivity to light and sound. Although initially linked to vascular alterations, findings in animal models have shown an important role in neuroinflammation [159]. In migraine patients, increased plasma levels of IL-10, TNFα, and IL-1 beta have been observed after a migraine attack and some studies have linked migraine attacks to local inflammatory events [160,161]. In a rat model of chronic migraine, PNU-282987 administration was found to relieve pain and decrease TNFα and IL-1β expression, and the calcitonin gene-related peptide. In addition, it reduced the number of astrocytes and microglia in the CA1 and CA3 regions of the hippocampus [162]. Thus, neuroinflammation could play an important role in the onset of symptoms.

In 2018, the FDA approved the use of VNS as a treatment for migraine attacks, although the beneficial effect is slight. Several studies suggest that VNS may help in the acute treatment of migraine by decreasing pain intensity, and mild adverse effects were reported [163,164]. It is also targeted for use in preventive treatment for episodic migraine, as observed in a Clinical Trial with 477 patients, where nVNS reduced the mean number of migraine episodes per month, relative to the group receiving stimulation alone [165]. However, the mechanism by which nVNS may decrease migraine symptoms is so far unknown.

Boström et al. studied cytokine levels in 12 migraine patients treated with nVNS, finding a decrease in pain along with an increase in IL-1 β and oxytocin in saliva [166]. In another study, serum cytokine levels were evaluated in 48 migraine patients treated with nVNS. The authors reported low plasma IL-1β levels compared to sham treatment [98]. Thus, both studies suggest that inflammation may be a therapeutic target in the treatment of migraine and the positive effect of nVNS in migraine patients could be due to its generalized anti-inflammatory properties.

### 4.7. Schizophrenia

Several reports have also linked neuroinflammation to schizophrenia [167]. For example, patients with schizophrenia show increased leukocyte counts [168] and increased levels of proinflammatory cytokines, such as IL-6, TNF-α, IL-1β, IL-12, and TGF-β [169,170,171]. Postmortem studies in patients with schizophrenia show increased microglia activation in the hippocampus and gray matter [172]. Hence, a possible modulation of inflammation through the cholinergic anti-inflammatory reflex has been proposed as a therapeutic target against schizophrenia [5].

Various α7 nAChR agonists have been evaluated in the treatment of schizophrenia due to their effects on cognition [173] and some have shown encouraging results in preclinical and phase 1 and 2 clinical studies. For example, in murine models of schizophrenia, chronic nicotine treatment reverses hypofrontality-impairment of cognitive processes critically dependent on the prefrontal cortex [174]. In addition, the agent 3-(2,4-dimethoxybenzylidene) anabaseine (DMXB-A), an α7 nAChR partial agonist, ameliorates negative symptoms in patients with schizophrenia [175], and Encenicline (EVP-6124), an α7 nAChR partial agonist and serotonergic 5-HT3 antagonist (5-hydroxytryptamine), showed promising results in preclinical testing and phase 1 and 2 clinical trials [176].

Although early results showed the usefulness of α7 nAChR agonists, in improving cognition, there are still no approved therapies. This is due to the reported side effects and lack of efficacy in subsequent clinical trials. For example, in a clinical trial in phase 2, an extended-release formulation of DMXB-A was tested and no improvement in cognition or clinical manifestations was found [177]. Furthermore, in a clinical trial in phase 3, which included more than 1500 patients with schizophrenia, Encenicline treatment showed no improvement in cognitive dysfunctions [176]. A possible explanation for these results is that the studies included tobacco-smoking patients in whom the plasma drug level is usually lower due to increased metabolism [177]. In addition, possible desensitization of nicotinic receptors must be considered.

Terry Jr & Callahan (2020) and Tregellas & Wylie (2019) have extensively reviewed the α7 nAChR agonists that have been tested in the treatment of cognition disturbances in schizophrenia [173,176]. Although no positive results have been found in phase 3 clinical trials there is still a need to test different doses of the agonists, as lower doses showed the best results, which is in agreement with the desensitization of nicotinic receptors at high doses. It is also important to mention that the study of α7 nAChR agonists in schizophrenia focused mainly on cognition, with no reports evaluating their effect on inflammation.

The use of VNS as a therapy against schizophrenia was explored in an animal model of mice treated with methylozoxymethanol acetate, where VNS prevented the increased psychomotor response to amphetamine, a consistent manifestation in patients with schizophrenia and murine models. This effect was related to VNS the reversal of hyperactivity within ventral hippocampal regions and the aberrant function of mesolimbic dopaminergic neurons [178]. However, a study conducted on 20 patients with schizophrenia and subjected to a Transcutaneous VNS protocol showed no improvement in the clinical manifestations of schizophrenia [179]. So far cholinergic modulation has been extensively studied in schizophrenia. However, studies have focused on its effects on cognition, hence studies evaluating improvements in neuroinflammation are required, as modulation of inflammation in schizophrenia as a therapeutic target is a promising field [167].

## 5. Blockade of the Anti-Inflammatory Cholinergic Pathway Mediated by MicroRNAs

An interesting strategy to regulate the activity of the anti-inflammatory cholinergic reflex is the repression of the protein translation involved in this pathway through microRNAs (miRNAs) and transfer RNA fragments (tRFs) [180]. Both miRNAs and tRFs are small non-coding RNAs that form complexes by complementarity, cleaving target messenger RNAs (mRNAs), repressing or blocking their translation [181,182]. MiRNAs have been shown to modulate the translation of proteins involved in the synthesis, secretion, and degradation of acetylcholine, such as choline acetyltransferase, vesicular acetylcholine transporter, AChE, and BChE [183,184].

It is known that miR-132 is overexpressed by leukocytes during the inflammatory process, leading to repression of acetylcholinesterase expression and attenuating inflammation through the anti-inflammatory cholinergic pathway. A recent study revealed that transgenic mice missing the miR-132 binding site and expressing acetylcholinesterase exhibited increased levels of inflammatory mediators (IL-1β, Il-6, and IL-10) and cholinergic pathway alterations [185]. Furthermore, in patients with inflammatory bowel disease, an increase in miR-132 also reduced acetylcholinesterase activity leading to amelioration of inflammatory state [186]. Besides, in EAE mice, miR-132 expression attenuates the disease, suppresses T-cell proliferation, and decreases IL-17 and IFN-γ production [187].

Other miRNAs such as miR-124, miR-199a, and miR-186, have been shown to act on the cholinergic system. MiR-124 inhibits intestinal inflammation by blocking the expression of AChE and STAT-3 [188]. In addition, in LPS-treated macrophages, miR-124 reduces the production of IL-6 and TNF-α converting enzymes (TACE) [189]. On the other hand, miRNA-199a and miR-186 decreased the expression of cholinesterases thereby favoring the cholinergic anti-inflammatory pathway and suppressing the expression of inflammatory cytokines [183,184].

Walgrave et al. (2021) have reviewed the possible application of miRNAs for the treatment of AD. Their work proposed the treatment of AD based on miRNAs; however, clinical trials are still required. To date, in vitro and animal models have shown that alternative treatment miRNAs-based is a viable strategy to ameliorate the inflammation [190]. Finally, the tRFs have been associated with cholinergic blockade of the immune response after stroke [191]. Therefore, the modulation of the cholinergic anti-inflammatory pathway by miRNAs is a promising alternative; nevertheless, further preclinical and clinical studies are needed.

## 6. Discussion

The nervous system and the immune system maintain close bidirectional communication under physiological and pathological conditions. Various processes in the CNS have a direct effect on the immune response such as pain, fear, and stress. One of the most studied examples of the connection between the brain and the immune system is the inflammatory reflex mediated by the vagus nerve, which is characterized by an anti-inflammatory effect. In this manuscript we review the information on how the inflammatory reflex can be approached for the treatment of various pathologies of the nervous system, focusing on pharmacological modulation and electrical modulation by vagus nerve stimulation.

Chemical stimulation with cholinergic agonists has shown beneficial results in reducing inflammation in EAE, Parkinson’s disease, epilepsy, depression, migraine, and schizophrenia. In addition, it has other beneficial effects such as improving cognitive ability in AD, reducing pain in migraines, and reducing depression. However, some studies show benefits when the cholinergic pathway is inhibited, as in EAE and depression. The differences in the results of chemical modulation of the cholinergic pathway are related to the model where it is evaluated, the type of agonist used, and the dose of the agonist. All neuronal nAChRs become temporarily inactive after prolonged exposure to an agonist, so receptor desensitization is an important point that impacts treatment doses.

The articles reviewed show information on the beneficial effects of VNS as an adjuvant treatment in diseases of the central nervous system such as depression and epilepsy, as well as in pathologies where its use as a potential therapeutic strategy is recently being explored, such as AD and PD. The mechanism of action of VNS is not fully elucidated and has been observed to have both afferent and efferent effects. While some studies attribute the beneficial role of VNS to the direct effect on CNS, others attribute its therapeutic usefulness to the activation of the anti-inflammatory reflex. Several clinical trials, which showed the effectiveness of VNS treatment on CNS diseases, did not evaluate the effect on the immune response. Therefore, it would be interesting to include in all future studies an analysis of the overall inflammatory profile. With the implementation of non-invasive VNS (tVNS and aVNS) allowing greater acceptability among patients, further studies could be performed to optimize the parameters of width, frequency, current intensity and type of stimulus, etc., to provide a more specific therapy for each disease and subgroup of people. 

## 7. Conclusions

In recent years the importance of the immune response in neurological and neurodegenerative diseases has been remarked, and considering the studies in both animal models and clinical research discussed here, it might be proposed that the modulation of the immune response in neurological and neurodegenerative diseases could be an alternative therapeutic approach.

## Figures and Tables

**Figure 1 ijms-22-13427-f001:**
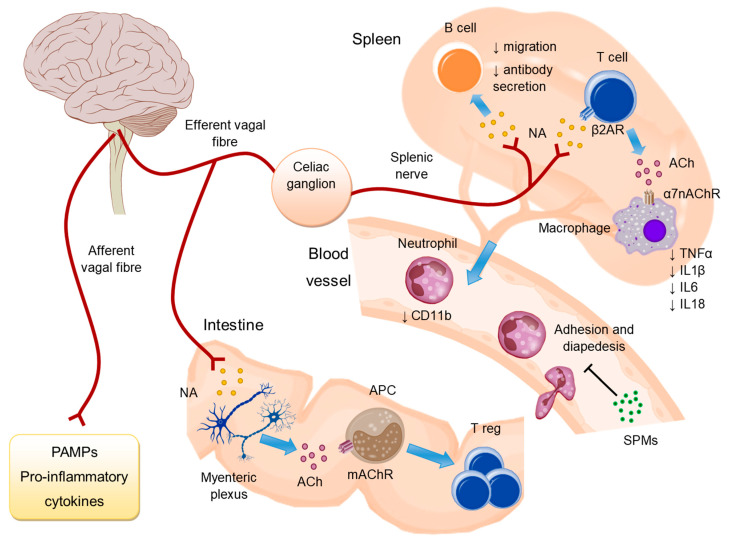
In the anti-inflammatory reflex, the afferent vagus nerve responds to inflammatory mediators and sends signals to the brainstem, where a signal originates and is transmitted by the efferent vagus nerve to the splenic nerve. This causes the release of noradrenaline (NA) in the spleen. CD4+ T lymphocytes expressing the beta-2 adrenaline receptor (β2AR) uptake NA and release acetylcholine (ACh). Acetylcholine inhibits the production of proinflammatory cytokines in macrophages expressing the α7nAChR receptor (α7 nicotinic acetylcholine receptors). In addition, the anti-inflammatory reflex reduces CD11b expression on neutrophils, stimulates the release of pro-resolving mediators (SPMs), and decreases antibody secretion and migration of B lymphocytes. In the intestine, ACh stimulates antigen-presenting cells (APCs), through muscarinic receptors (mAChR), to favor the maintenance of regulatory T cells (Treg).

**Table 1 ijms-22-13427-t001:** VNS parameters tested in clinical trials of central nervous system diseases.

Disease	Range of Parameters	Approvals	Inflammatory Profile	Refs
Human Epilepsy	0.75–1.75 mA, 30 Hz	FDA	no significant changes	[95,96]
Human Depression	0.25 mA, 25 Hz	FDA	non evaluated	[97]
Human Migraine	0.25 mA, 25 Hz	FDA	↓IL-1β plasma levels	[98]
Parkinson’s disease	maximum output of 60 mA, 25 Hz	not approved	↓ TNF- α	[99]
Alzheimer’s disease	0.4–0.8 mA, 20 Hz	not approved	non evaluated	[92,93]

VNS, Vagus nerve electrical stimulation; FDA, Food and Drug Administration; Hz, Hertz; mA, milliampere; TNF-α, Tumour Necrosis Factor-α; IL-1 β, interleukin-1 β.

## Data Availability

Not applicable.

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
