# Peer review of "Role of the Cholinergic Anti-Inflammatory Reflex in Central Nervous System Diseases"

_ijms, 2021, doi:10.3390/ijms222413427_

Round 1
Reviewer 1 Report
This interesting and comprehensive review covers the research evidence supporting the concept of the cholinergic blockade of inflammation and its translational implications. The text is exceptionally well written, which is a strength point of this manuscript; however, it only deals with the protein regulators involved in the intriguing blockade process, which in these days of RNA-therapeutics, and given that ample evidence is available on RNA modulators of the cholinergic blockade process, is a considerable weakness. The authors are encouraged to add text and accompanying bibliography on the RNA mediators of the cholinergic blockade of inflammation.
Author Response
Answer to reviewer comments
Reviewer 1
This interesting and comprehensive review covers the research evidence supporting the concept of the cholinergic blockade of inflammation and its translational implications. The text is exceptionally well written, which is a strength point of this manuscript; however, it only deals with the protein regulators involved in the intriguing blockade process, which in these days of RNA-therapeutics, and given that ample evidence is available on RNA modulators of the cholinergic blockade process, is a considerable weakness. The authors are encouraged to add text and accompanying bibliography on the RNA mediators of the cholinergic blockade of inflammation.
Answer: A new section on RNA mediators of the anti-inflammatory reflex has been added, including new references. Lines 666-699
Reviewer 2 Report
The manuscript “Role of the cholinergic anti-inflammatory reflex in central nervous system diseases” review current data supporting positive effects that stimulation cholinergic anti-inflammatory reflex stimulation show, or may show, in the treatment of various diseases in the CNS. The manuscript is well structured and data are reported in an appropriate sequence.
Although the text is understandable, editing by a native English speaker is needed to comply to IJMS standards.
A high number of oversights also should be corrected, some of them are listed in the minor issue section, however, full name, family, and biological activity of each bioactive molecule should be provided every time they are cited. The same goes for experimental settings and animal models of studies that are described.
major issue
Lines 215-216: although the articles are correctly cited, the idea that amyloid fibrils might have a protective effect is, at least, “controversial” and authors should provide a more detailed discussion of this issue.
Ls 495-500: how do this sentence correlate with epilepsy. If the correlation is indirect authors should remove it from this chapter and shift it in a more proper one.
minor issues
line 45-46. authors should state here which medical practice, involving the anti-inflammatory reflex are applied in therapies against CNS diseases.
line 68: “ injecting low doses of CNI-1493 intracerebroventricularly; a p38 MAPK” …. do authors agree that: “intrecerebroventricular injections of CNI-1493 (a p38 MAPK…..)” could be easier to understand?
line 77: replace” tractus solitaries” with “tractus solitarius”
Lines 81-82: the cited paper do not mention adrenaline
Lines 88-89: please specify the experimental model
Lines 89-90: please detail the experimental model
Lines 102-103: please specify the animal model everywhere they have been omitted
Line 162: typo?
Lines 200-204: these data should be supported by a citation of the original text rather than of another review review citing it.
Lines 239-241: detail the experimental setting
Lines 367-370: how do cytoprotective effects of α7 nAChRs correlate with the anti-inflammatory reflex?
Lines 371-377: authors should correlate these sentences with the anti-inflammatory reflex, or eliminate them
Line 383: authors should specify here what PNU-282987 is
Line 520: authors should provide extended name of the HPA axis
Line 578…what authors mean with “surrounding levels of IL-10?
Lines 1095-1097: this bibliographic entry is incorrect
Round 2
Reviewer 1 Report
Thanks for the satisfactory revision.
Reviewer 2 Report
Authors response to almost all of my issues were exhaustive. I still think that English prose in a review should be more elaborated than in an original research article, and I was i little bit disappointed they did not follow my suggestion. However, as I said it was just a suggestion to increase the quality of a paper that provide lot of informations in a well structured sequence.